

# Genome-wide identification and characterization of the KCS gene family in sorghum (*Sorghum bicolor* (L.) Moench)

Aixia Zhang[1], Jingjing Xu[1], Xin Xu[1], Junping Wu[2], Ping Li[1], Baohua Wang[1] and Hui Fang[1]

[1] Ministry of Agricultural Scientific Observing and Experimental Station of Maize in Plain Area of Southern Region, School of Life Sciences, Nantong University, Nantong, Jiangsu, China
[2] Nantong Changjiang Seed Co., Ltd, Nantong, Jiangsu, China

## ABSTRACT

The aboveground parts of plants are covered with cuticle, a hydrophobic layer composed of cutin polyester and cuticular wax that can protect plants from various environmental stresses. $\beta$-Ketoacyl-CoA synthase (KCS) is the key rate-limiting enzyme in plant wax synthesis. Although the properties of *KCS* family genes have been investigated in many plant species, the understanding of this gene family in sorghum is still limited. Here, a total of 25 *SbKCS* genes were identified in the sorghum genome, which were named from *SbKCS1* to *SbKCS25*. Evolutionary analysis among different species divided the *KCS* family into five subfamilies and the SbKCSs were more closely related to maize, implying a closer evolutionary relationship between sorghum and maize. All *SbKCS* genes were located on chromosomes 1, 2, 3, 4, 5, 6, 9 and 10, respectively, while Chr 1 and Chr 10 contained more *KCS* genes than other chromosomes. The prediction results of subcellular localization showed that SbKCSs were mainly expressed in the plasma membrane and mitochondria. Gene structure analysis revealed that there was 0–1 intron in the sorghum *KCS* family and *SbKCSs* within the same subgroup were similar. Multiple *cis*-acting elements related to abiotic stress, light and hormone response were enriched in the promoters of *SbKCS* genes, which indicated the functional diversity among these genes. The three-dimensional structure analysis showed that a compact spherical space structure was formed by various secondary bonds to maintain the stability of SbKCS proteins, which was necessary for their biological activity. qRT-PCR results revealed that nine randomly selected *SbKCS* genes expressed differently under drought and salt treatments, among which *SbKCS8* showed the greatest fold of expression difference at 12 h after drought and salt stresses, which suggested that the *SbKCS* genes played a potential role in abiotic stress responses. Taken together, these results provided an insight into investigating the functions of *KCS* family in sorghum and in response to abiotic stress.

Corresponding authors
Baohua Wang, bhwang@ntu.edu.cn
Hui Fang, fanghui8912@126.com

## INTRODUCTION

In higher plants, very long chain fatty acids (VLCFAs) are precursors for the synthesis of waxes in pollen husks, fruits, stems and leaves (*Paul et al., 2006*). In addition, VLCFAs exist in plants in the forms of several other substances, including triacylglycerol, suberin, phospholipids and sphingolipids (*Haslam & Kunst, 2013*). VLCFAs are synthesized by extending C16 or C18 fatty acids through microsomal enzyme system. They participate in a variety of cellular processes such as nuclear pore formation, and lipid and protein transport (*Paul et al., 2006*). Plant cuticular wax has important physiological functions in resisting abiotic stresses, such as preventing water loss, resisting ultraviolet radiation, drought and saline alkali stress (*Lee & Suh, 2015a*; *Lee & Suh, 2015b*). Therefore, VLCFAs play an important role in cell biology, whereas many aspects of their synthesis and function remains to be further studied.

Plant cuticular waxes have a very complex chemical composition, mainly including VLCFAs and alkanes, primary alcohols, secondary alcohols, aldehydes, ketones and esters (*Bernard & Joubès, 2013*). The synthesis and secretion of waxes need the participation of a variety of organelles and enzymes. The synthesis of VLCFA is catalyzed by four enzymes, including $\beta$-ketoyl-CoA synthase (KCS), $\beta$-ketoyl-CoA reductase (KCR), $\beta$-hydroxyacyl-CoA dehydratase (HCD) and enol-CoA reductase (ECR). Among them, KCS has substrate specificity and tissue specificity, and is a key enzyme in the elongation process of VLCFA (*Fan et al., 2018*). It participates in the synthesis of wax precursors and prolongs VLCFAs in higher plants, thereby limiting non-stomatal water loss, protecting against UV radiation, resisting bacterial and fungal pathogen invasion, and defending against drought stress (*Zhang et al., 2019*). Because of the significance of *KCS* genes, their functions have been well studied and understood in some plant species. Three *KCS* genes, including *KCS1* (*Todd, Post-Beittenmiller & Jaworski, 1999*), *FDH* (*Pruitt et al., 2000*) and *CER6* (*Hooker, Millar & Kunst, 2002*), had been proved to be involved in the biosynthesis of cuticular waxes in Arabidopsis. *FDH* was mainly expressed in the young leaves and flower organs of Arabidopsis and related to the synthesis of VLCFAs in epidermal cells, regulating the formation of plant morphological structure and improving the resistance to biotic or abiotic stresses (*Pruitt et al., 2000*). *KCS2/DAISY* is a wax biosynthesis gene in Arabidopsis that is activated by direct binding to *MYB94* (*Lee & Suh, 2015b*) and *MYB96* (*Seo et al., 2011*). An increase in the accumulation of cuticular wax decreased the rate of cuticular transpiration in the leaves of *MYB94*-OE lines, thus improving the drought resistance of plants (*Lee & Suh, 2015b*). Overexpression of *KCS* in wheat significantly increased the resistance to adversity of wheat (*Hu et al., 2010*). In barley, the mutation of *KCS1* gene affected the wax structure of stratum corneum, thereby reducing the resistance to water and powdery mildew of cuticles (*Li et al., 2018*). Therefore, *KCS* genes played an important role in the synthesis of wax components and in the response to stress environments in plants. Besides the function in response to adverse stresses, the *KCS* genes have potential to improve some quality traits, e.g., *KCS13* could promote the increase of fiber length in cotton (*Qin et al., 2007*). Currently, *KCS* genes had been identified in several species. For example, 21 *KCS* genes in Arabidopsis were identified and classified into four subgroups,

namely *FAE1*-like, *KCS1*-like, *FDH*-like and *CER6* (*Costaglioli et al., 2005*). In addition, 58 *KCS* genes in *Gossypium hirsutum* (*Xiao et al., 2016*), 28 *KCS* genes in *Malus domestica* (*Lian et al., 2020*), 31 and eight *KCS* genes in *Brassica rapa* and *Picea abies* (*Guo et al., 2016*), respectively, have been reported.

Sorghum (*Sorghum bicolor* (L.) Moench), one of the earliest cultivated cereal crops in China, has great edible value and is rich in nutrients with a lot of fat-soluble protein and unsaturated fatty acids. The *KCS* gene family plays a crucial role in the synthesis of wax and VLCFAs in many plants, but little is known about the *KCS* gene family in sorghum. Therefore, in this study, we systematically identified and analyzed the characteristics of sorghum *KCS* gene family by bioinformatics methods, and the expression levels of *SbKCS* genes under drought and salt stresses were also investigated. This study provided useful information for further investigating the molecular functions of *KCS* genes in response to abiotic stress in sorghum.

## MATERIALS AND METHODS

### Identification of *SbKCS* genes in sorghum

The sorghum BTx623 genome dataset was downloaded from the Ensembl Genomes website (http://ensemblgenomes.org/). The hidden Markov model (HMM) profiles of the 3-Oxoacyl-[acyl-carrier-protein (ACP)] synthase III C terminal domain (ACP_syn_III_C, Pfam:PF08541) and the FAE1/Type III polyketide synthase-like protein domain (FAE1_CUT1_RppA, Pfam:PF08392) were obtained from the Pfam protein family database (http://pfam.xfam.org/) (*Liu et al., 2018*). Firstly, after using the Hmmsearch program in the Linux system to search for proteins containing the conserved domain (*Jeanmougin et al., 1998*), more than 50 protein sequences were obtained with expected (E) values less than 1.2E-28. Then, to ensure the accuracy and reliability of the detected proteins, NCBI-CDD website (https://www.ncbi.nlm.nih.gov/cdd), Pfam and SMART databases (http://smart.embl-heidelberg.de/) were used to further detect the KCS-conserved domains in the non-redundant sequences of each putative protein to confirm these sequences (*Mistry et al., 2021*). Finally, 25 *SbKCS* genes were identified and named according to their positions on chromosomes. The online server PROTPARAM (https://web.expasy.org/protparam/) was used to analyze the theoretical isoelectric point (Pi), molecular weight (MW), and sequence length of all KCS proteins. CELLO website (http://cello.life.nctu.edu.tw/) was used to predict the subcellular localization of the KCS proteins by submitting the sequences of SbKCS proteins.

### Chromosomal location and gene structure analysis of *SbKCS* genes

The physical locations of all *SbKCS* genes were obtained in the sorghum genome. The MapChart (*Voorrips, 2002*) and AI (Adobe Illustrator CS6) software were used to visualize the images of chromosomal position. The online program of the gene structure display server (GSDS: http://gsds.cbi.pku.edu.cn) was used to obtain the exon-intron structure of the *SbKCS* genes according to their full-length CDS sequences.

## Multiple sequence alignment of SbKCS proteins

To analyze the conserved domains of proteins, the multiple sequence alignment of 25 identified SbKCS proteins and 21 Arabidopsis KCS proteins was performed using Clustal W in MEGA 7.0 (*Kumar, Stecher & Tamura, 2016*) and then the result of multiple sequence alignment was imported into GeneDoc website (http://nrbsc.org/gfx/genedoc) to visualize the conserved sequences of the SbKCS proteins.

## Phylogenetic and motif analysis of *SbKCS* genes

To study the phylogenetic relationships of SbKCS proteins, the sequences of KCS proteins in Arabidopsis, sorghum, maize, rice and *Brochypodium distachyon* were aligned with Clustal W software under its default parameters. The phylogenetic tree was constructed using the maximum-likelihood (ML) method of MEGA 7.0 software with 1,000 bootstrap replicates. Evolview online website (https://evolgenius.info/evolview-v2/#login) and AI software were used to visualize the phylogenetic tree. An online MEME program (http://meme-suite.org/) was used to analyze the conserved motifs in the identified sorghum KCS proteins and the parameter settings were as follows: the optimum width of motif was between 6 and 50 residues, and maximum number of motifs was 10. TBtools software was used to visualize the conserved motifs based on output XML files (*Chen et al., 2020*).

## Secondary structure prediction and three-dimensional (3D) model construction of SbKCS proteins

Secondary structure of SbKCS proteins was predicted using an online website Prabi (https://npsa-prabi.ibcp.fr). To further analyze the protein structure of the SbKCS family, the tertiary structure of SbKCS proteins was predicted according to the protein sequences. Then, the online tools SWISS-MODEL (https://swissmodel.expasy.org/) was used to construct the 3D models of SbKCS proteins by homologous protein modeling method (*Waterhouse et al., 2018*).

## Prediction of *cis*-acting elements of *SbKCS* genes

The 1.5 kb promoter sequences upstream of the initiation codon for all the 25 *SbKCS* genes were extracted from sorghum genomic sequences, and then they were analyzed by PlantCARE (http://bioinformatics.psb.ugent.be/webtools/plantcare/html/) (*Wang et al., 2020a*; *Wang et al., 2020b*) to survey the potential *cis*-acting elements associated with stress response and hormones in the promoter regions. The obtained *cis*-element information was then visualized through the GSDS online website.

## Plant materials and stress treatments

The seeds of sorghum variety BTx623, an elite inbred line whose genome have been sequenced (*Paterson et al., 2009*), were planted in pots filled with Pindstrup substrate and vermiculite (3:1) and cultivated in a controlled and adjusted green house at a temperature of 23 °C/20 °C with a 16 h/8 h day and night regime. After growing to the three-leaf stage, sorghum seedlings with consistent growth were selected and fixed on the planting basket, and cultured with 1/2 Hoagland nutrient solution (pH 5.5). Two different abiotic stresses were applied to the seedlings at the three-leaf stage. For salt stress, the sorghum seedlings

were cultured in 150 mM NaCl solution for 24 h (*Wei et al., 2021*); for drought stress, the seedlings were hydroponically cultured with 20% PEG6000 nutrient solution for 24 h (*Wang et al., 2020a*; *Wang et al., 2020b*); the control group was cultured with 1/2 Hoagland nutrient solution. The sorghum seedlings under stresses and control were sampled at different time after treatment for 0, 6, 12 and 24 h, respectively. Three biological replicates for each sample were collected and quickly put into liquid nitrogen, and then stored in an ultralow temperature freezer −80 °C for RNA extraction.

## RNA extraction, and quantitative real-time PCR (qRT-PCR) analysis

Total RNA was extracted from leaves at different stages under stress treatment and control using Ultrapure RNA Kit (CWBio, Jiangsu, China). About 1 µg of total RNA was used to synthesize the first-strand cDNA using the HiScript III RT SuperMix for qPCR kit purchased from Nanjing Vazyme Company (Nanjing, China). The qPCR primers were designed with Primer 5.0 software and the primer sequences were shown in Table S1. The qRT-PCR was carried out for each sample using a ChamQ SYBR qPCR Master Mix (Vazyme, Nanjing, China) with an ABI 7500 real-time PCR system and the parameters were set as follows: stage 1, 95 °C for 30 s; stage 2, 40 cycles of 95 °C for 10 s and 60 °C for 30 s; stage 3, melting curve. The *actin1* (SORBI_3001G112600) was used as an internal reference for standardization between samples (*Zheng et al., 2021*). For qRT-PCR detection, three biological replicates and four technological replicates for a single sample were performed to ensure the accuracy of qPCR analysis. Relative expression levels of each gene were calculated using the $2^{-\Delta\Delta CT}$ method (*Livak & Schmittgen, 2001*). The data were presented as mean ± standard error (SE) and Student's *t* test was used for significance analysis, and $P < 0.05$ was considered as significant. The histogram was drawn with GraphPad Prism software (*Swift, 1997*).

## RESULTS

### Identification of *KCS* genes in sorghum

More than 50 putative *KCS* genes were obtained by searching two conserved domains. Subsequently, the protein sequences of the putative *SbKCS* genes were confirmed through SMART, NCBI-CDD websites and Pfam databases, and some incomplete and uncertain sequences were removed after screening. Ultimately, a total of 25 *SbKCS* genes were identified as belonging to the *KCS* gene family and were named as *SbKCS1-SbKCS25* based on their location on the sorghum chromosomes. Furthermore, the physicochemical properties of the SbKCSs were characterized (Table S2). The SbKCS proteins with largest and smallest number of amino acids were SbKCS5 and SbKCS21, containing 549 and 440 amino acids, respectively. The molecular weight of these proteins was in the range of 48785.3 (SbKCS21) −60617.4 Da (SbKCS5), and the theoretical isoelectric point ranged from 7.1 (SbKCS17) to 9.91 (SbKCS9), with a mean value of 8.92.

### Chromosomal location and subcellular localization prediction analysis of *SbKCS* genes

To map the chromosomal location of the *SbKCS* genes, the physical location of the *SbKCS* genes on the chromosomes was investigated. Each *SbKCS* gene was named in the light

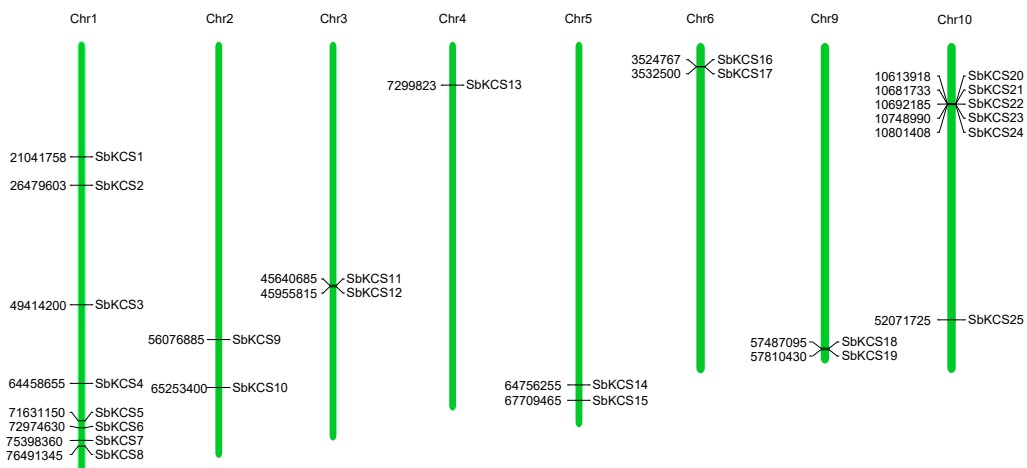

**Figure 1** **Chromosomal localization of the *SbKCS* genes.** 25 *SbKCS* genes were mapped on eight sorghum chromosomes. The chromosome numbers were indicated at the top of each vertical green bar. The gene names on each chromosome corresponded to the approximate locations of each *SbKCS* genes. The number shown to the left of each vertical green bar was the specific physical position of each *SbKCS* gene on the chromosome.

of its top-down physical location on sorghum chromosome (Chr) 1 to 10 and the 25 *SbKCS* genes were unevenly distributed on the chromosomes (Fig. 1). *SbKCS* genes were distributed separately or in clusters, and mostly at both ends of chromosomes. Chr 1 had the largest number of *SbKCS* genes (8, ~32.0%), followed by Chr 10 (6, ~24.0%); Chr 4 possessed the least *SbKCS* genes (1, ~4.0%). Each of the other chromosomes (Chr 2, 3, 5, 6, 9) contained 2 (~8.0%) *SbKCS* genes. The subcellular localization prediction analysis showed that SbKCS proteins were mainly localized in the plasma membrane, followed by mitochondria and chloroplast, which suggested that SbKCS proteins might be mostly expressed and function in these organelles (Fig. 2).

## Mutiple sequence alignment, phylogenetic and conserved motifs analysis

To evaluate phylogenetic relationship among the members of the *KCS* family in sorghum, Clustal W in MEGA 7.0 software was used to perform multiple sequence alignment for the 25 identified SbKCS and 21 AtKCS protein sequences. The results showed that it contained two important conserved domains: the FAE1_CUT1_RppA domain and the ACP_syn_III_C domain (Fig. S1). The sequences of these two domains were highly consistent and conserved in both regions, indicating that they were essential to the function of *KCS* genes in sorghum. Meanwhile, we also found that the C-terminal and intermediate regions of these proteins were highly conserved, whereas the N-terminal was poorly conserved.

To clarify the phylogenetic relationship among sorghum KCS proteins and obtain more detailed classification of the KCS proteins, we constructed a phylogenetic tree using the MEGA 7.0 based on the full-length protein sequences of the 25 obtained SbKCS and KCS proteins from the representative species: Arabidopsis (21), rice (19), maize (29) and

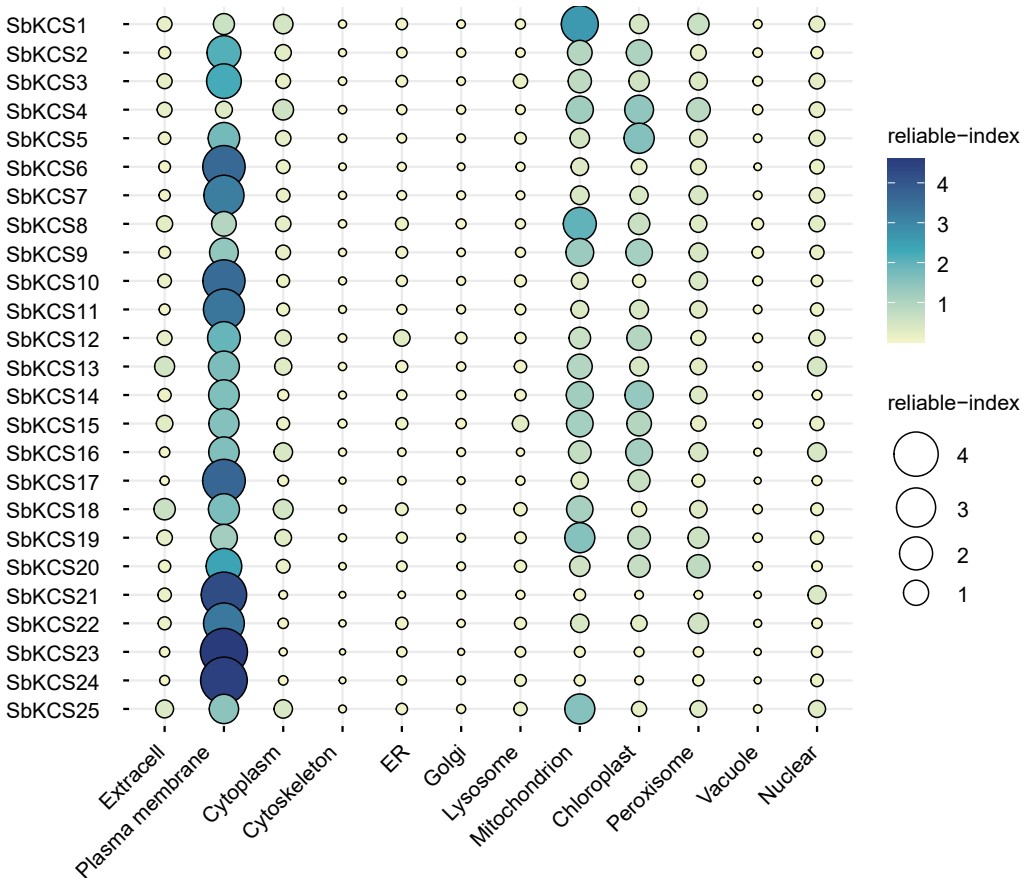

**Figure 2** **The prediction of subcellular localization for SbKCS proteins.** The color and the size of the circles indicated the reliability of the prediction results. The name of each protein was shown on the left. The site name for the predicted subcellular localization of each SbKCS protein was shown at the bottom.

*Brochypodium distachyon* (32) (Fig. 3). Subsequently, the 126 KCS proteins were divided into five subfamilies named from Group 1 to Group 5. SbKCS proteins were distributed in all five groups. Group 1 had the largest number (10) of SbKCS members, followed by Group 4 with the number of 9. Group 5 contained 4 members of the KCS family. Besides, Group 2 and Group 3 each contained only 1 SbKCS protein. In previous studies, 21 AtKCS proteins were classified into 4 subfamilies, namely FAE1-like, CER6, KCS1-like, and FDH-like (*Costaglioli et al., 2005*). Similarly, we also found that 9 *SbKCSs* in Group 1 appeared in the KCS1-like subfamily and the remaining one gene in Group 1, *SbKCS18*, was clustered in FAE1-like subfamily; 9 *SbKCSs* in Group 4 appeared in the CER6 subfamily; 4 *SbKCSs* in Group 5 appeared in the FDH-like subfamily. The remaining two *SbKCSs*, *SbKCS6* and *SbKCS7*, were branched separately into two new branches, Group 2 and Group 3. These results indicated that the evolutionary relationship of *KCS* family genes in sorghum was more complex than that in Arabidopsis. Furthermore, KCS proteins in sorghum and maize were always in close branches in all subgroups, which implied that the SbKCSs were more closely related to maize in evolutionary relationship. In addition, genes from the same

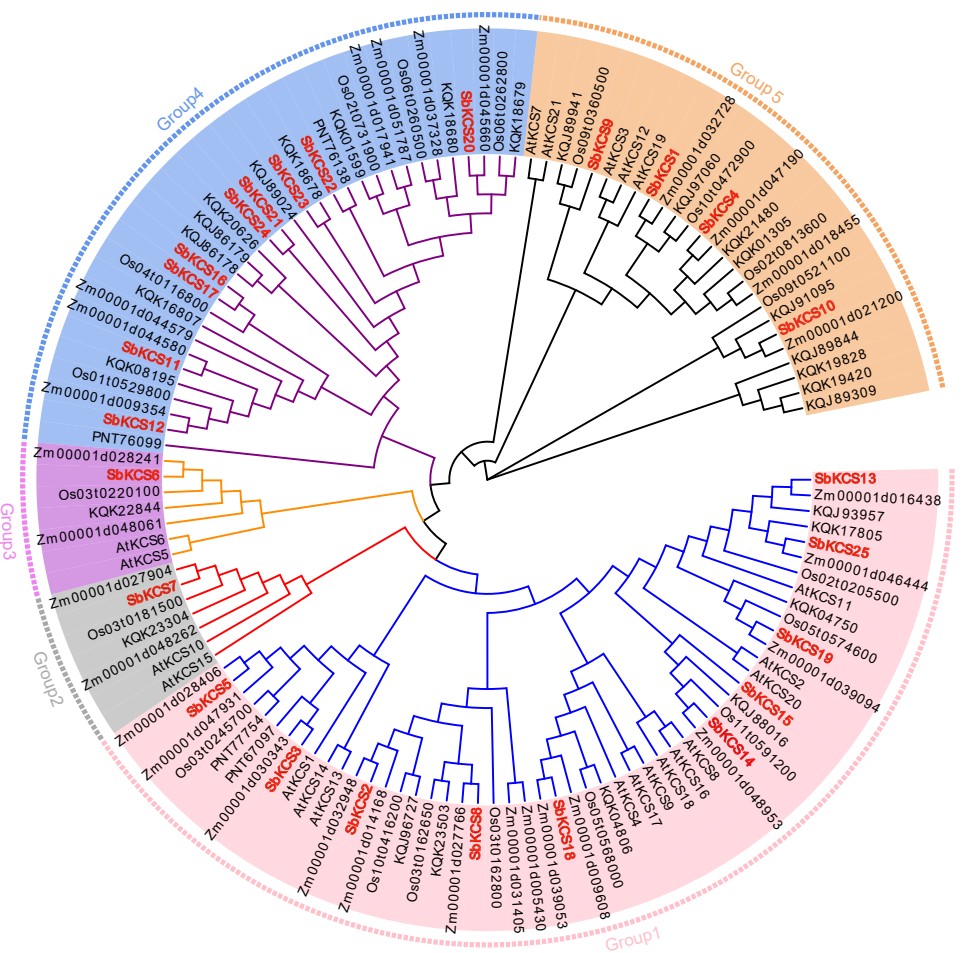

**Figure 3** **Phylogenetic tree of KCS gene family members in five species.** The full-length protein sequences from Arabidopsis, sorghum, maize, rice and *Brachypodium distachyon* were used to construct the phylogenetic tree by MEGA 7.0 software based on a maximum likelihood (ML) method with bootstrap replicates of 1,000. Five groups were highlighted in different colors. The members of KCS family in sorghum were marked in red.

subgroup could be thought to have similar functions; consequently, the five distinct clades might indicate that the functions of *KCS* gene were diverse.

In order to obtain the characteristic regions of SbKCS proteins, the analysis of conserved motifs was performed using MEME software. Ten motifs in the SbKCS proteins were predicted and finally, 5 distinct conserved motifs were found (Fig. 4A). Each of the 25 SbKCS proteins contained motifs 1, 2, 4, 6 and 8, with the same order that they were arranged on the protein sequences, indicating that the motifs of almost all the identified *KCS* family genes were conserved.

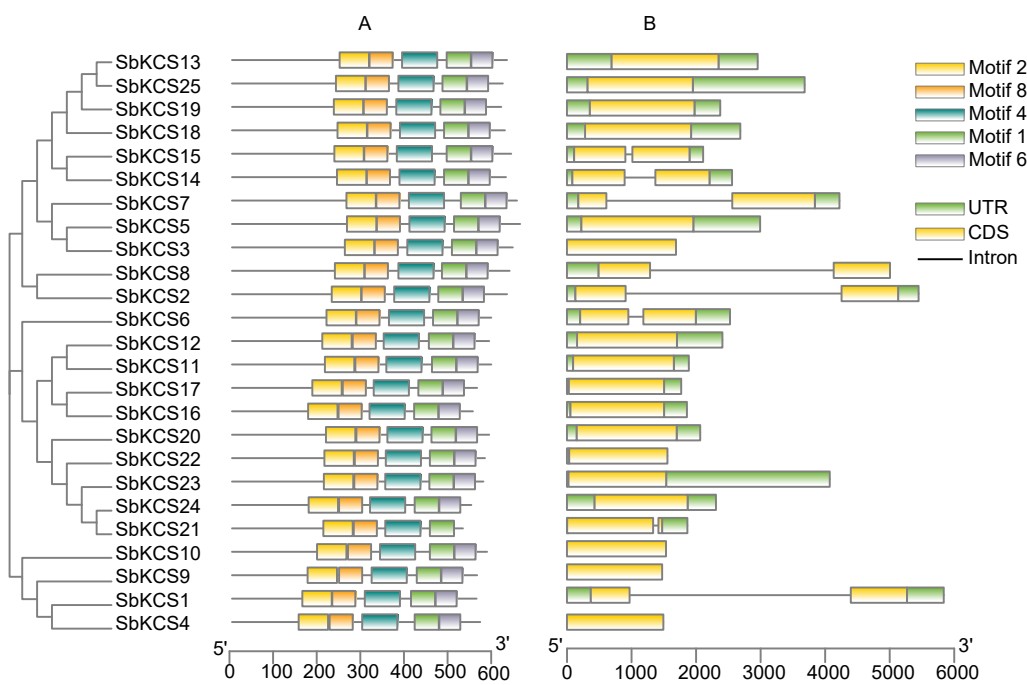

**Figure 4 The motif and gene structure analysis of the KCS family members in sorghum.** (A) Motif analysis of the 25 SbKCS proteins. The conserved motifs in the SbKCS proteins were identified with MEME software. The grey lines denoted the non-conserved sequences, and each motif was showed by a different color. The length of motifs in each protein was presented in proportion. (B) The exon-intron structure analysis of the 25 *SbKCS* genes. Exons were represented by yellow rectangles, introns were represented by gray lines, and green rectangles represented UTR regions.

## Gene structure, secondary structure and 3D modelling analysis of the SbKCS proteins

To analyze the structural composition of each *SbKCS* gene, the exon-intron organizations were obtained by comparing the cDNA and DNA sequences of a single *SbKCS* gene (Fig. 4B). From the results, we discovered that most of the *SbKCS* genes were relatively simple in structure, e.g., the number of introns in the 25 *SbKCS* genes ranged from 0 to 1. Most of the *SbKCS* genes (17, ~68.0%) contained no introns in their open reading frame regions, and 8 (~32.0%) *SbKCS* genes contained only 1 intron. The 17 intronless genes were distributed in Group 1, 4 and 5, while the *SbKCS* genes containing only one intron were distributed in all the five subgroups. As shown in Fig. 4B, the exon-intron structure of members within the same subgroup tended to be similar. The number of *SbKCS* genes with and without introns in Group 1 was about the same. In Group 1, four genes including *SbKCS2*, *SbKCS8*, *SbKCS14* and *SbKCS15* each had one intron, whereas the remaining six *SbKCS* genes had no intron. All the *SbKCS* genes in Group 2 and 3 have no introns. In Groups 4 and 5, only one gene in each group contained one intron, and the others had no introns. Generally speaking, genes in the same subfamily contained similar genetic structures and the position of introns tended to be conserved in each subgroup. These results further illustrated the

authenticity of the evolutionary tree, and it was speculated that the intron-exon structure of these *SbKCS* genes was also related to the evolution of the *KCS* gene family.

In view of the fact that the structure and function of proteins were closely related, to have a clearer understanding of the function of *SbKCS* genes, we predicted their secondary structure and constructed a tertiary structure model of all SbKCS proteins. We found that alpha helices, extended strands, beta turns and random coils constituted the secondary structure of SbKCS proteins (Table S3), of which the alpha helix (40.84%–50.00%) accounted for the largest proportion, followed by random coils (32.12%–37.84%) and extended strands (11.99%–16.49%), and the beta turns (4.46%−7.51%) accounted for the smallest proportion. In addition, from the tertiary structure of SbKCS proteins constructed by the homology modeling (Fig. 5), we could see that the SbKCS proteins had similar 3D structures, and the composition and position of secondary structure of the SbKCS proteins could be clearly observed. The $\alpha$-helix, random coils, $\beta$-turn and extended strands that made up the secondary structure of SbKCS proteins were further coiled and folded through the interaction of side chain groups, and a compact spherical space structure was formed by the maintenance of various secondary bonds. The stability of tertiary structure was necessary for proteins to exert biological activity.

### *Cis*-acting elements analysis of the *SbKCS* gene promoters

To further study the underlying molecular function of *SbKCS* genes, the 1.5 kb promoter sequences of *SbKCS* genes were extracted, and the PlantCARE online site was used to analyze the possible *cis*-acting regulation elements (CAREs). A total of 10 *cis*-acting elements related to abiotic stress response and hormones were found, including two stress-related elements (W-box and MBS) and eight hormone-related elements (STRE, ABRE, I-box, MYB, G-Box, Gap-box, TCT-motif and GA-motif) (Fig. 6). The ABRE related to the abscisic acid response was detected in 19 *SbKCS* genes and the STRE, involved in the salicylic acid response, was revealed in 20 *SbKCS* genes. Furthermore, 11 and 14 *SbKCS* genes contained W-box and MBS elements, which were involved in pathogen trauma response and drought-induced processes, respectively. The *cis*-acting elements of G-box, TCT-motif, GA-motif, Gap-box and I-box related to light responsiveness were identified in 20, 9, 4, 2 and 5 *SbKCS* genes, respectively. Besides, the *cis*-acting element MYB was identified in 24 *SbKCS* genes, which played a role in controlling plant secondary metabolism (*Uimari & Strommer, 1997*), regulating cell morphogenesis (*Yang et al., 2007*) and environmental factor responses (*Lea et al., 2007*). These results showed that the *SbKCS* genes might play a role in hormone signaling pathways and biotic or abiotic stress responses.

### Expression analysis of *SbKCS* genes under drought and salt stresses

Previous researches have shown that *KCS* genes are involved in the response to plant abiotic stresses (drought and salt) (*Lee & Suh, 2015a*; *Lee & Suh, 2015b*) and plant hormones (ABA) (*Macková et al., 2013*). The identification of *cis*-acting elements related to abiotic stress response in the promoter region of *SbKCS* genes also indicated their potential functions in response to different abiotic stresses. Consequently, to further determine the potential molecular function of *SbKCS* genes in response to abiotic stresses in sorghum,

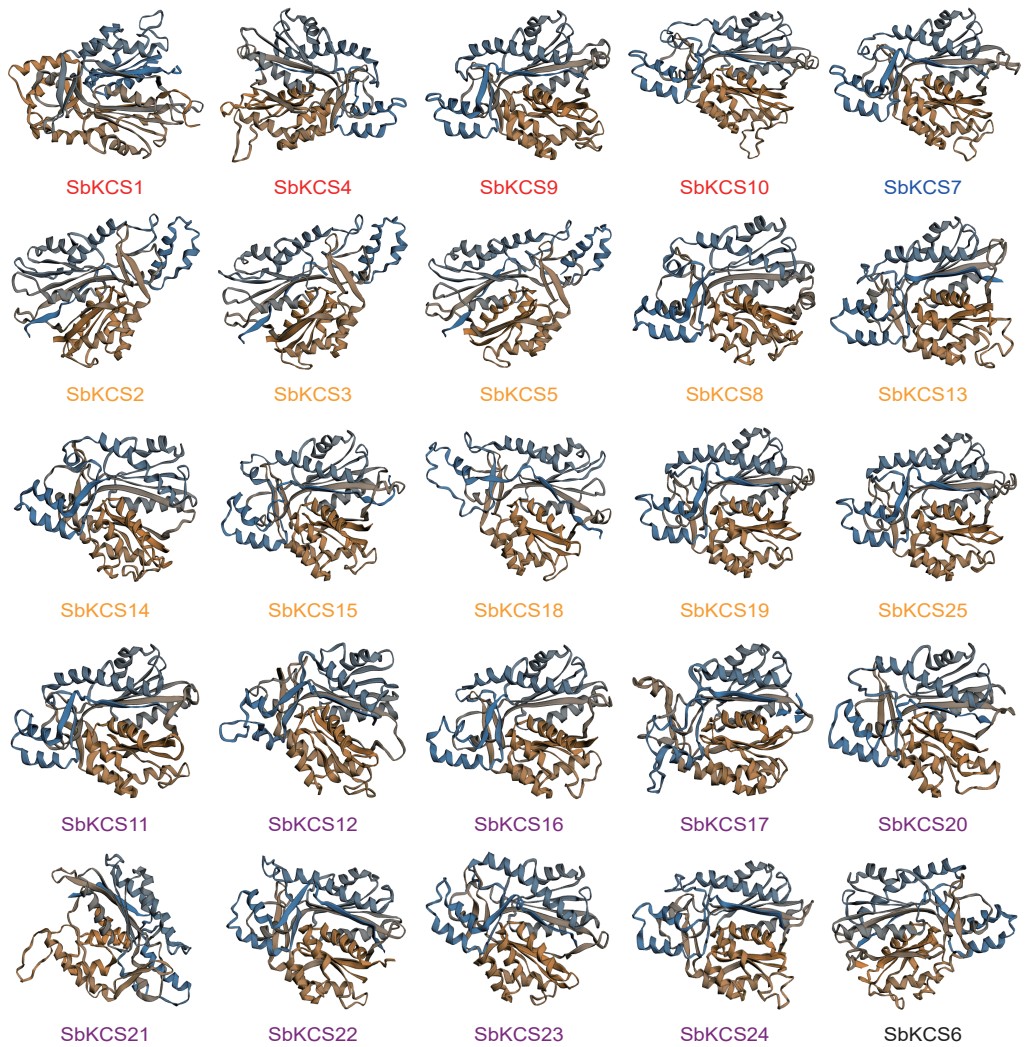

**Figure 5** **Three-dimensional (3D) models of the SbKCS proteins.** The 3D model of SbKCS protein was constructed by using the method of homologous protein modeling on SWISS-MODEL online website. Proteins of the different subfamily were shown in different colors at the bottom of each 3D model. Orange fonts represented the Group 1 subfamily; blue fonts represented the Group 2 subfamily; black fonts represent the Group 3 subfamily; purple fonts represented the Group 4 subfamily and red fonts represented the Group 5 subfamily.

a total of nine *SbKCS* genes were randomly selected from the five subgroups, and their expression levels at 0, 6, 12 and 24 h after drought and salt treatments were analyzed (Table S4). The results showed that nine randomly selected *SbKCS* genes were up-regulated or down-regulated within 24 h under drought and salt stresses. Under drought stress, the expression of 7 genes except *SbKCS6* and *SbKCS8* were up-regulated at different times after treatment (Fig. 7). The expression level of *SbKCS3* was significantly down-regulated at 24 h after treatment (Fig. 7B). In addition, the expression levels of *SbKCS1*, *SbKCS3*, *SbKCS14* and *SbKCS16* significantly increased at 6 h after drought treatment, suggesting that these

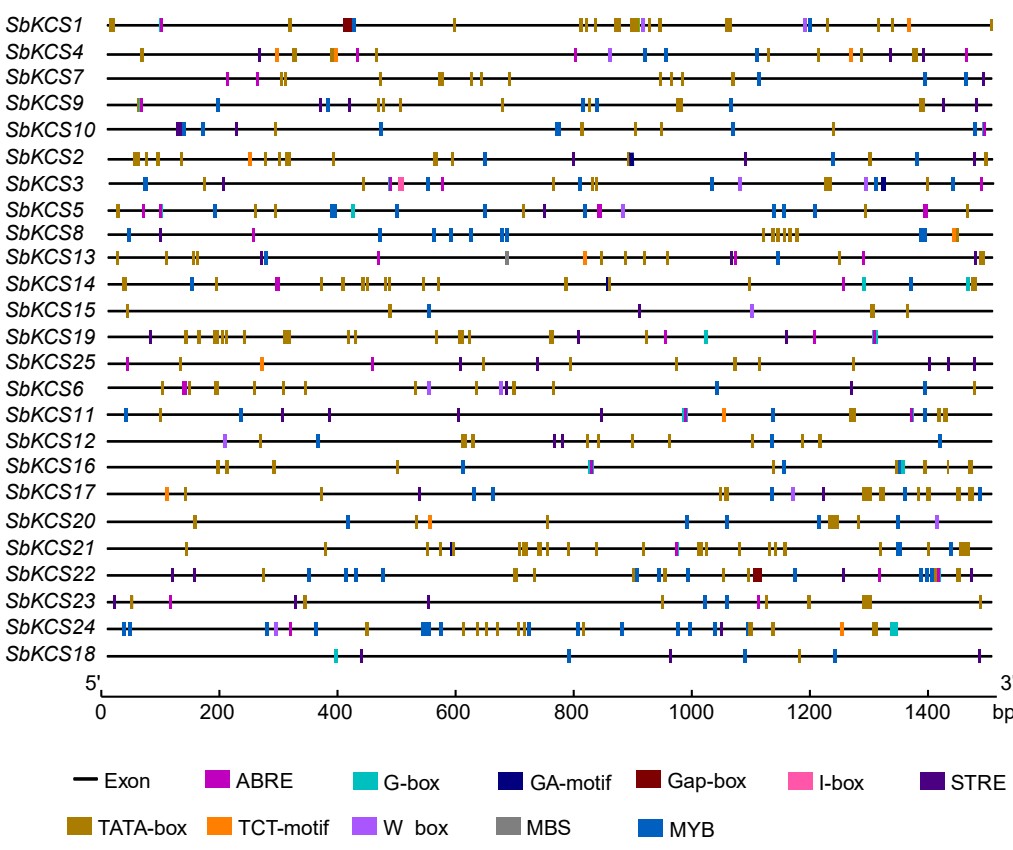

**Figure 6** *Cis-acting elements in the SbKCS gene promoters.* The different *cis*-acting elements were shown in different colors. The name of each gene was shown on the left. Promoter sequence length was displayed in proportion.

*SbKCS* genes could rapidly respond to drought stress. Furthermore, *SbKCS14* and *SbKCS18* showed a continuous response at 6–12 h and 12–24 h after drought treatment, indicating the key role in drought resistance (Figs. 7F and 7H).

Under salt treatment, the expression levels of the nine selected *SbKCS* genes except *SbKCS7* showed significant differences at different time after treatment compared to the control (Fig. 8). *SbKCS3* and *SbKCS6* showed similar expression pattern under salt treatment, in which the expression levels of the two genes increased firstly at 6 h after salt treatment and then declined (Figs. 8B and 8C), suggesting the complex of their function in response to salt stress. In addition, *SbKCS14*, *SbKCS16* and *SbKCS18* were significantly up-regulated at 6 h or 12 h after salt treatment (Figs. 8F and 8H), and *SbKCS1*, *SbKCS8* and *SbKCS21* were down-regulated after treatment (Figs. 8A, 8E and 8I), of which the expression of *SbKCS8* declined significantly at both 6 h and 12 h under salt treatment. Interestingly, among all the differentially-expressed genes, no matter under drought or salt treatment, *SbKCS8* showed the largest differentially expressed fold change (at 12 h of drought and salt treatments), which implied that *SbKCS8* might be an important gene to resist drought and salt stresses in sorghum. Taken together, we found that different genes

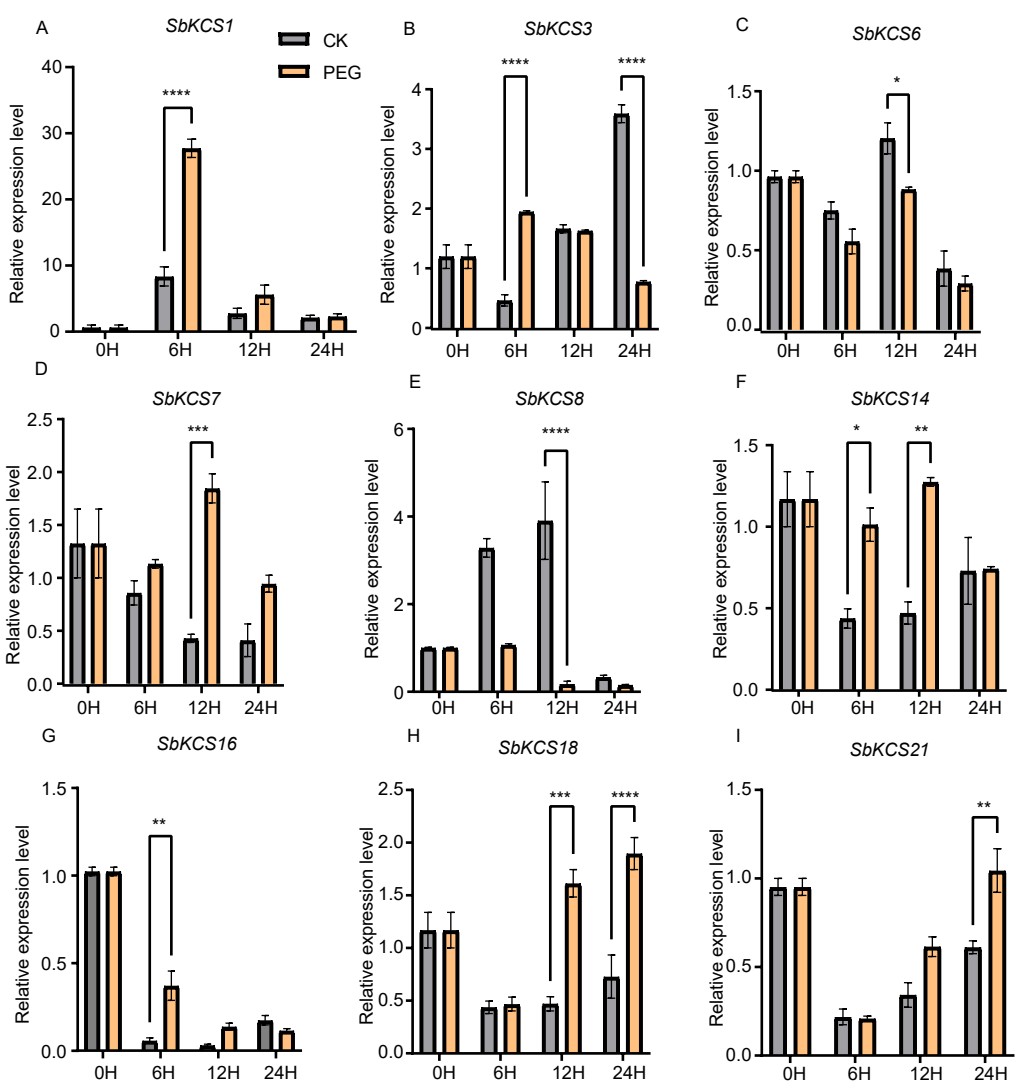

**Figure 7** **Expression profile of nine *SbKCS* genes under drought stress treatment.** The expression level of the nine *SbKCS* genes (A-I) were validated by qRT-PCR. Seedings were treated with drought (20% PEG6000) and leaves were sampled at 0, 6, 12 and 24 h. Data represent mean ± standard error (SE) of three biological replicates. Statistically significant differences between the control (CK) and treatment groups (PEG) are indicated using asterisks (* $p < 0.05$, ** $p < 0.01$, *** $P < 0.001$, **** $P < 0.0001$; independent Student's $t$-test).

showed different expression patterns under stress conditions and some of the *SbKCS* genes might be involved in response to abiotic stresses with different mechanisms.

# DISCUSSION

Cuticular waxes play a significant role in coping with abiotic stresses in plants and form an important protective barrier in the long-term ecological adaptation process of resisting harsh ecological environment, and biotic and abiotic stresses (*Bernard & Joubès, 2013*). In recent years, the functions of many genes related to wax synthesis have been widely

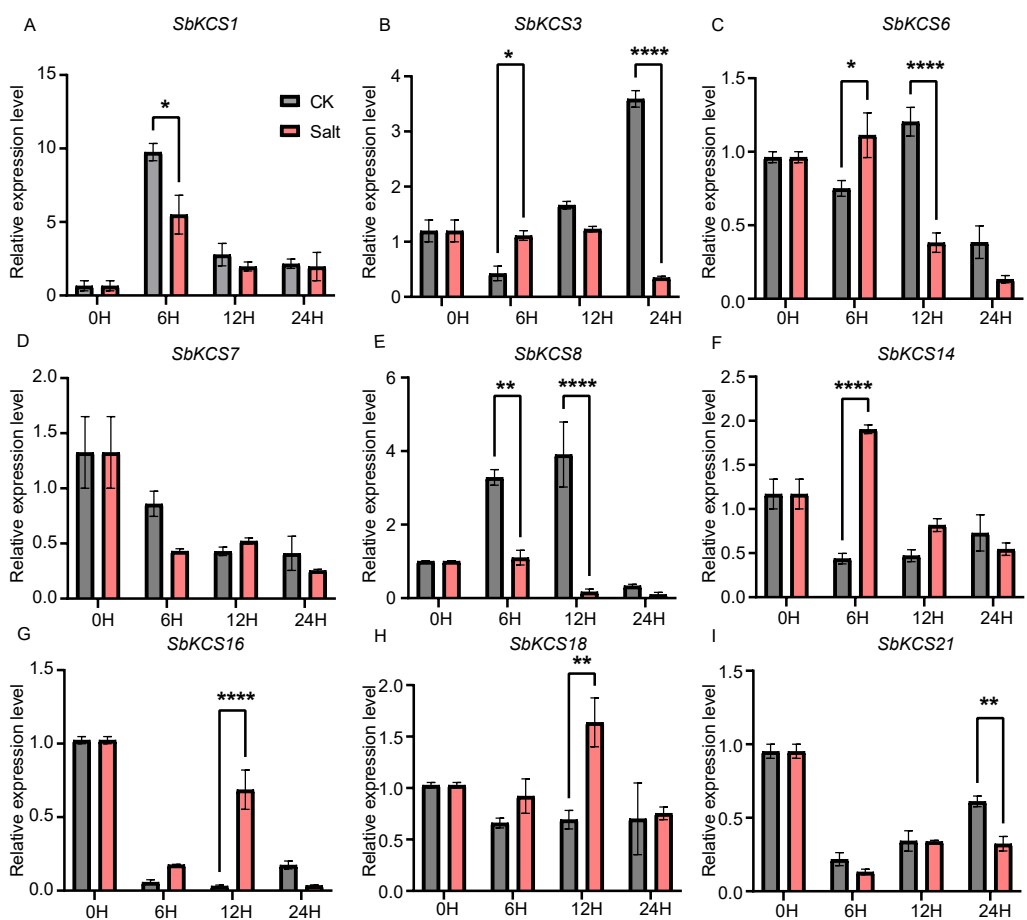

**Figure 8** **Expression profile of nine *SbKCS* genes under salt stress treatment.** The expression level of the nine *SbKCS* genes (A–I) were validated by qRT-PCR. Seedings were treated with salt (150 mM NaCl) and leaves were sampled at 0, 6, 12 and 24 h. Data represent mean ± standard error (SE) of three biological replicates. Statistically significant differences between the control (CK) and treatment groups (Salt) are indicated using asterisks (* $p < 0.05$, ** $p < 0.01$, *** $P < 0.001$, **** $P < 0.0001$; independent Student's *t*-test).

validated. For example, the *CER1* gene was related to the alkanes biosynthesis process and it was highly linked to responses to biotic and abiotic stresses (*Bourdenx et al., 2011*). The overexpression of *CER1* in Arabidopsis could increase the amounts of n-alkanes of chain lengths between 29 and 33 carbon atoms and the *cer1* mutant exhibited an altered n-alkanes biosynthesis (*Bourdenx et al., 2011*). CER1-LIKE1 in Arabidopsis could control the alkane biosynthesis and wax crystallization (*Pascal et al., 2019*). KCS was considered to be a key enzyme that determines the tissue and substrate specificities in the biochemical process of fatty acid elongation in higher plants (*Guo et al., 2020*). The biological functions of several *KCS* genes have been clarified by using some Arabidopsis mutants in previous studies. For example, *FAE1* cloned from Arabidopsis was shown to catalyze the biosynthesis of C20 and C22 VLCFA storing lipids in seeds and it was only expressed in seeds (*James Jr et al., 1995*). *CUT1/CER6* played a key role in the biosynthesis of cuticular wax C26 and longer

VLCFAs during epidermis and root hair development (*Pang et al., 2010*). Loss of *FDH1* function led to ectopic organ fusions, suggesting that *KCS* played a critical role in organ development (*Voisin et al., 2009*). Furthermore, complete loss of *DAISY*/*KCS2* function resulted in the accumulation of C20 VLCFA, which suggested that *KCSs* might be necessary for the biosynthesis of cuticular wax (*Franke et al., 2009*). So far, limited wax-related genes have been reported in sorghum, and the molecular and biological mechanism of wax synthesis and its response to abiotic stress of sorghum remain unclear.

In this study, 25 *SbKCS* genes were identified and characterized in sorghum, which could provide important information about the roles of *SbKCS* in plant's growth and development and abiotic stress resistance. The 25 *KCS* genes in sorghum contained the complete FAE1_CUT1_RppA and ACP_SYN_III_C domain. It was speculated that the function of *KCS* gene family was closely related to these two important conserved domains. The physicochemical properties of 25 SbKCS proteins was further calculated and it was found that they were weakly basic proteins with different isoelectric point. The discrepancies in structure and properties of the SbKCS proteins were presumed to adapt to the changes of the environment in the process of evolution. Lipid metabolism pathways are mainly located in plastid, endoplasmic reticulum (ER) and peroxisome of plants (*Li et al., 2013*). In previous studies, it was confirmed that some genes related to Arabidopsis wax synthesis were also expressed in the ER, such as *AtKCR1* and *AtKCR2* (*Beaudoin et al., 2009*), *AtKCS9* (*Kim et al., 2013*), *CER10* (*Haslam & Kunst, 2013*), and *CER1* (*Pascal et al., 2019*). However, most of SbKCS proteins were predicted to be mainly expressed in plasma membrane, mitochondria and chloroplast. The differences in subcellar location of KCS proteins in sorghum might indicate that these genes function in different ways.

Genes with similar gene structure and protein structure often have similar functions. The SbKCS proteins were revealed to be mainly classified into five distinct subgroups. Most of the *SbKCS* genes have no or only one intron in their open reading frame regions and the typical pattern of gene structure was in accord with those in Arabidopsis (*Joubès et al., 2008*) and Malus (*Lian et al., 2020*). In addition to further confirming the reliability of the *KCS* superfamily classification, these findings also suggested that the *KCS* superfamily genes had similar characteristics in dicots and monocots, which might lead to their functional similarity of the same families in different species. The conserved motifs of SbKCS proteins were also further analyzed in our study, and all SbKCS proteins shared the same motif composition and similar location distribution, suggesting that these proteins might have similar functions in plants. However, the research on these motifs is still limited, and their regulatory functions needs to be further explored.

The analysis of *SbKCS* gene chromosomal localization uncovered a gene cluster on Chr 10. The most common amplification method of gene families is replication (*Cannon et al., 2004*). It was speculated that the *KCS* gene family had undergone duplications during evolution. Interestingly, *SbKCS* genes were not found on Chr 7 or Chr 8. We speculated that this might be caused by chromosome shift or fragment loss in the process of evolution. Furthermore, the regulatory mechanism of each gene could be better understood by identifying the upstream *cis*-acting elements of *SbKCS* genes. *SbKCS* genes contained several *cis*-acting elements that could regulate gene expression by combining with *trans*-acting

factors. Genes containing different *cis*-acting elements might have different regulatory mechanisms. For instance, the G-box, a light-responsive *cis*-acting element, was involved in the regulation of plant flowering stage, which was greatly affected by light (*Menkens, Schindler & Cashmore, 1995*). ABRE, as a homeopathic element regulating seed and bud dormancy, could regulate stomatal switch and control gene expression via abscisic acid response, thus enhancing plant stress resistance (*Narusaka et al., 2003*). Genes containing MYB *cis*-acting element were speculated to play a role in the response to cold and drought resistance (*Shinozaki, Yamaguchi-Shinozaki & Seki, 2003*).

*KCS* gene family plays a significant role in determining the quantity and composition of VLCFAs, and the wax yield of aerial part of plants is affected by different environmental factors, such as light, moisture and temperature (*Shepherd & Wynne Griffiths, 2006*). Previous studies had revealed that *AtKCS6* transcript level was affected by changes in different environmental conditions, such as different light conditions or osmotic stress and these environmental signals promoted wax synthesis by up-regulating the expression of key enzymes in wax biosynthesis, thus improving plant stress resistance (*Hooker, Millar & Kunst, 2002*). Therefore, many studies have focused on the functional identification of *KCS* genes in plants, and the extensive response of *KCS* family genes to abiotic stress has been confirmed in many plants. In Arabidopsis, the expression levels of *KCS1*, *KCS3* and *KCS6* decreased under dark and low temperature conditions, but increased under osmotic tolerance (*Joubès et al., 2008*). The expression of *AtKCS6* decreased in dark environment, indicating that light was necessary for the expression of *AtKCS6* gene in stems and seedlings, while the expression of *AtKCS6* increased in stems and seedlings after NaCl and PEG treatment or dehydration treatment, which suggested its critical function in resistance to adverse environment (*Hooker, Millar & Kunst, 2002*). The expression of *MdKCS12* and *MdKCS24* in Malus was up-regulated 5 and 10 times, respectively, after 20% PEG treatment for 1 h and meanwhile, *MdKCS12* and *MdKCS24* could also be induced by salt stress (*Lian et al., 2020*). These results suggested that plants could respond to abiotic stress through regulating the expression levels of associated *KCS* genes. Therefore, to further study the potential functions of *SbKCS* genes in response to abiotic stress, the expression levels of nine randomly selected *SbKCS* genes were studied under drought and salt stress conditions by qRT-PCR. Most of the nine selected *SbKCS* genes were obviously induced or repressed when subjected to drought or salt stresses, clarifying that *SbKCSs* might play a role in response to drought and salt stresses in sorghum seedling. Meanwhile, we found that some *SbKCS* genes showed similar expression patterns under drought and salt treatments. For instance, the expression of *SbKCS8* was down-regulated under drought and salt treatments, whereas *SbKCS18* was up-regulated under these two stress treatments, which suggested that these two genes may respond to drought and salt stresses by a similar mechanism. Remarkably, the expression of *SbKCS1* was up-regulated at 6 h and 12 h after drought stress, but was down-regulated in the salt stress, indicating that *SbKCS1* might play different roles under two different stresses. The results further indicated the facticity of these genes and the possibility of their function in two stress responses. In brief, these results will provide useful information for the subsequent functional studies of *SbKCS* genes, and contribute to improve the stress resistance of sorghum germplasms.

## CONCLUSIONS

In this study, we identified 25 *KCS* genes in sorghum and classified them into five subgroups based on the phylogenetic tree, and the SbKCS and ZmKCS proteins showed a closer evolutionary relationship. These *SbKCS* family genes were unevenly distributed on eight chromosomes of sorghum. The analysis of their conserved motif compositions and gene structures showed high levels of consistency and similarity within the same subgroup. *Cis*-acting element analysis of these identified *SbKCS* genes contributed to a further study of their performance in plant growth and stress responses. The gene expression levels of *SbKCS* genes in leaves under drought and salt stresses were also analyzed by qRT-PCR, which hinted at their potential roles in response to abiotic stresses. The results found in this study provided a theoretical basis for investigating the function of *KCSs* in sorghum and also provided valuable gene resources for the application of wax-related genes to improve stress resistance and genetic characteristics of sorghum.

### Funding

This work was supported by the Social Livelihood Science and Technology Project of Nantong City, China (MS22020033), the Nantong University Scientific Research Start-up project for Introducing Talents (135420609055), the Practice Innovation Training Program Projects for College Students in 2021 (202110304081Y) and the Large Instruments Open Foundation of Nantong University. The funders had no role in study design, data collection and analysis, decision to publish, or preparation of the manuscript.

### Grant Disclosures

The following grant information was disclosed by the authors:
Social Livelihood Science and Technology Project of Nantong City, China: MS22020033.
Nantong University Scientific Research Start-up project for Introducing Talents: 135420609055.
Practice Innovation Training Program Projects for College Students in 2021: 202110304081Y.
Large Instruments Open Foundation of Nantong University.

### Competing Interests

Junping Wu is employed by Nantong Changjiang Seed Co., Ltd.

### Author Contributions

- Aixia Zhang performed the experiments, analyzed the data, prepared figures and/or tables, authored or reviewed drafts of the article, and approved the final draft.
- Jingjing Xu performed the experiments, analyzed the data, prepared figures and/or tables, and approved the final draft.
- Xin Xu analyzed the data, prepared figures and/or tables, and approved the final draft.

- Junping Wu analyzed the data, prepared figures and/or tables, and approved the final draft.
- Ping Li conceived and designed the experiments, authored or reviewed drafts of the article, and approved the final draft.
- Baohua Wang conceived and designed the experiments, authored or reviewed drafts of the article, and approved the final draft.
- Hui Fang conceived and designed the experiments, authored or reviewed drafts of the article, and approved the final draft.

## Data Availability

The raw data are available in the Supplemental Tables.

## Supplemental Information

Supplemental information for this article can be found online at http://dx.doi.org/10.7717/peerj.14156#supplemental-information.

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
