# Peer review of "Genome-wide identification and characterization of the KCS gene family in sorghum (Sorghum bicolor (L.) Moench)"

_PeerJ, doi:10.7717/peerj.14156_

## Round 0.1 · original submission · Major Revisions

The article is not acceptable in its present form. Much effort is required to improve this manuscript. The scientific writing also has to be improved as it is not up to the standards. Kindly incorporate carefully all the points as suggested by reviewers.

Reviewer 1 ·

Basic reporting

Authors have identified 25 genes encoding Keto-acyl Co-A synthase in sorghum using bioinformatic tools. They identified the chromosome location, conserved domains and motifs of these 25 SbKCS proteins. In order to validate the function of some of these genes in response to drought and saline stress, they have treated the seedlings of Sorghum with PEG and NaCl for different durations and performed qRT-PCR analysis. They observed differential expression of the tested genes in response to stress treatments.

Authors have written the manuscript well. They have provided relevant details in the introduction section, described the results appropriately and provided all the necessary raw data files for the review.
However, the manuscript needs a brief grammatical and language check before getting accepted for publication.

Experimental design

Experimental design is appropriate for the study and enough replicates were included to obtain statistically significant results.
However, in the material and methods section authors need to provide little more details about how the stress treatments were conducted. Additional comments and suggestions are detailed in the edited version of the manuscript.

Validity of the findings

No comments

Annotated reviews are not available for download in order to protect the identity of reviewers who chose to remain anonymous.

·

Basic reporting

Authors of this work have identified a total of 25 SbKCS genes based on the sorghum genome, named SbKCS1 to SbKCS25 and divided into four 24 subgroups, including KCS1-like, FAE1-like, FDH-like, and CER6. They have further characterized physicochemical properties, chromosome distribution, subcellular localization, conserved motifs, phylogenetic tree, gene structure, secondary and tertiary structures, and cis-acting elements of SbKCS genes. Following are my comments.

1) Please rearrange method section sub-headings,with a clear focus on bioinformatics methods and then experimental parts

2) Number of species in phylogenetic analysis is less; you can add a few more KCS family members from other plant species like rice, Brachypodium, maize, and wheat

3) Re-analyse the phylogenetic tree with the best-predicted model with the maximum likelihood method and support the tree with bootstrap

4) Predicted 3-D model of KCS members needs to be evaluated using various available programs example – the swiss model assessment tool.

4) Please write in detail all figure headings and description

Experimental design

1) Please rearrange method section sub-headings,with a clear focus on bioinformatics methods and then experimental parts

2) Number of species in phylogenetic analysis is less; you can add a few more KCS family members from other plant species like rice, Brachypodium, maize, and wheat

3) Re-analyse the phylogenetic tree with the best-predicted model with the maximum likelihood method and support the tree with bootstrap

Validity of the findings

Good; Predicted 3-D model of KCS members can be evaluated using various available programs example – the swiss model assessment tool.

Additional comments

NONE

Reviewer 3 ·

Basic reporting

The quality of Scientific writing is not up to the mark in the manuscript (MS), so does the quality of English language. There are many unambiguous statements all through the MS. There are improper uses of singular and plurals in the text. There are also few spelling mistakes, and improper use of adverbs and adjectives. There are improper and incomplete sentences also in the text. It is not possible to comment here for all of those. Authors should consider rewriting to improve the Science and English language of the MS.

Although the structure of article is alright, but figures are not professionally depicted. Structures of the 25 SbKCSs are put in one figure (Figure 5) without a proper explanation about the purpose of doing it. qRT-PCR figures are put for each of the 9 genes tested in the study (figure 7).

Although the bioinformatics analyses and gene expression data looks alright, but the conclusion depicted in the MS by the authors for those data are overstated. Please, go through my additional comments for examples and details.

Experimental design

Authors did not pursue a rigorous investigation about the KCSs genes in Sorghum. The whole study requires more investigation, and possibly more relevant data.

There are also certain improper understanding about background of the research topics according to the way INTRODUCTION is written in the manuscript.

Materials and Methods sections are quite brief about the tools and techniques used in the study.

Validity of the findings

There are new findings in the study, but the study requires to investigate more and have more data to make a better conclusion according to the aim of the study. Conclusion in the study based on the available data are often overstated.

Additional comments

The quality of Scientific writing is not up to the mark in the manuscript (MS), so does the quality of English language. Authors wrote, ‘induced to be upregulated’ (line 264), ‘progressively clarified’ (line 294), prolonging C16 or C18 (line 38), formation of epidermal (line 56,57) - these kinds of unscientific uses of words are consistently written in all through text of the MS. Authors mentioned ‘waxy’ in four places (line 51,66, 64,60), which is supposed to be ‘wax’. Those are not, at all, typo error. Authors required to understand the difference between 'wax' and 'waxy'.
There are improper uses of singular and plurals in the text. There are also few spelling mistakes, and improper use of adverbs and adjectives. There are improper and incomplete sentences also in the text. It is not possible to comment here for all of those. Authors should consider rewriting to improve the Science and English language of the MS.
There is also inconsistency about the Scientific ways of writing the names of organisms, genes and mutants. For an example, if authors like to put ‘Arabidopsis’ as italic (the genus name of Arabidopsis thaliana), then the sorghum (Sorghum bicolor) name supposed to be mentioned in italic as ‘Sorghum’, and so does the apple (Malus domestica) name suppose to be ‘Malus’ (in italic). This inconsistency exists in all through the manuscript (MS), and required to be fixed.
It is also preferable to put gene or gene family name in italic, for example, instead of putting FAE1-like, or KCS1-like (please check line 70-71 in the PDF version of the MS), authors should consider writing FAE1 (italic)-like and KCS1 (italic)-like. Also, instead of putting SbKCS genes, authors should consider using SbKCS (italic) genes. In my opinion those need to be changed in all through the text of the MS to keep the consistency.
Line 289 ‘Cer1’ need to be changed to ‘cer1’. So, ‘fae1’ need to be changed to ‘fae1’. ‘cis’-acting and ‘trans’-acting need to be cis (italic)-acting and trans (italic)-acting – the other italic issues need to be fixed all through text of the MS.
I have also many scientific arguments and concerns about many sections of the MS.
The author mentioned in the abstract and elsewhere (as a conclusion) that their results suggest that the SbKCS genes play a significant role in the abiotic stress responses, and indicated a close relationship between fatty acids and abiotic stress resistance in Sorghum. I think it is too much of an overstatement from the data they have in the study. They identified, bioinformatically, 25 SbKCS genes from Sorghum, and in turn, they did some expression studies of 9 of those 25 genes in response to drought and salinity induced by PEG and NaCl. There are no detail gene functional and physiological studies reported in the manuscript other than the simple expression studies. There are even no data about changes in the wax biosynthesis in Sorghum by the up- or down-regulation of any identified SbKCS gene. Without reporting proper functional studies, authors need to minimize their claims about their findings.
Authors mentioned in the first paragraph of the INTRODUCTION that ‘VLCFAs are key components of waxes in pollen husks, leaves and some plant seeds’. In contrary to this statement, the popular understanding is that cuticular waxes are basically VLCFAs and their derivatives, such as, VLC primary and secondary alcohols, VLC aldehydes, VLC alkanes and VLC ketones. Cuticular waxes can be found in the cuticle of whole aerial sections of the plants, which include both leaves and stems. Authors need to aware of the existence of stem cuticle and stem cuticular waxes. Therefore, author should consider rewriting the confusing and misleading sentences by exploring more into the literature of plant cuticle and cuticular wax.

In the first paragraph, authors also mentioned ‘epidermal wax’ (line 40). In plants, waxes are accumulated in the cuticle of whole aerials parts and are called cuticular waxes. There are both intracuticular and epicuticular waxes in most plant cuticle. What does the authors indicate about ‘epidermal wax’? Do cuticular waxes in plants are also called epidermal waxes? I doubt that.

Authors mentioned about BTx623 in the Materials and Method (M&M) section? Is it the variety/cultivar name of Sorghum they used as plant material? Authors need to mention a bit detail about BTx623.

Authors also mentioned in the Plant material of M&M that Sorghum was grown in the greenhouse at a temperature of 23 ℃ with a 16 h/8 h day and night regime. Is it a temperature-controlled greenhouse? How did the authors control the day-light length (16h/8h) in a Greenhouse?

‘ACTIN’ is a housekeeping gene (line:154). Proper naming is required. How did authors identified the SbACTIN gene from Sorghum? Also, qRT-PCR gene analysis supposed to include more than one housekeeping gene to normalize the expression of the target genes. Also, the level of ACTIN expression is normally very high in plant tissues. Target genes with low basal expression in plan tissues are not supposed to be normalized with ACTIN in quantitative PCR analysis. A low-expression house-keeping control required to be chosen.
How did authors decide about those 50 putative KCS genes from Sorghum? How did they distract 25 out of 50? Need more elaboration description for their methodology in the M&M section. It is not clear in the methods and also in the result sections.

In overalls, most sections of the methods are quite brief. More descriptions are required.

Why using of Arabidopsis KCSs for phylogenetic analysis? Sorghum is a C4 plant and a monocot. It would be interesting to see the homology comparison of SbKCSs with Zea mays (corn) KCSs. Like Sorghum, Zea mays is also a monocot and C4 plant species. However, Sorghum, as a species, is more drought resistant than Zea mays. So, its worthy to generate the phylogenetic tree with other monocots to have a better understanding about the evolutionary aspect of KCSs in Sorghum.

Regarding the protein structure, authors need to put a much better and precise description of KCS enzyme proteins in plants. How are the alpha helices are intertwined with random coils, extended strands and beta-turns? Putting only the percentage information of each structural components is not interesting and not enough description of the KCS protein structure.

What did the authors really mean by regulation of plant abiotic stresses (line 255-66)? KCSs are structural protein genes, not regulatory genes. Expression of genes under any abiotic stress is kind of a response. It’s not the regulation of the stress.

In figure 7, authors do need to put 9 individual figures for showing the KCS gene expression data in response to PEG and NaCl treatments. One figure for PEG and one figure for NaCl displaying the expressions of 9 genes together should be looked better and more informative. There is no way to know the expression levels of each target genes as compared to others. If all of them put together in one bar graph with a comparative level of expression, that will be good.
In the paragraph of line 271-282, what does it mean by upregulation and downregulation of KCSs in response to drought and salinity? Do the up- and down-regulation of KCS gene expressions change the biosynthesis or accumulation of wax constituents in the cuticle? If so, authors should consider wax extraction from Sorghum cuticle and analyze by Gas Chromatography-Mass Spectroscopy (GC-MS).
Line 289-290: Did cer1 mutant showed alteration in paraffin biosynthesis? Are the authors indicating alkane biosynthesis? Are alkanes also called paraffins? Do plants produce paraffins? It’s a wrong use of reference. I went through the Bernard et al. (2012), but nothing about the role of CER1 protein in paraffin biosynthesis.

Annotated reviews are not available for download in order to protect the identity of reviewers who chose to remain anonymous.

---

## Round 0.2 · Major Revisions

The article is not acceptable in its present form. Still much effort is required to improve this manuscript. The scientific writing is not up to the standards. Kindly incorporate carefully all the points as suggested by the reviewer.

·

Basic reporting

Authors have revised their manuscript based on suggestions.
1. I feel the manuscript needs more re-writing, right from the abstract to conclusions. English usage is still far from satisfactory upto the journal standards. For ex. phrases like theoretical and molecular basis
extensively participate, resistance of wheat to adversity, weakening the water resistance, synthesis of VLCFA is transformed by four distinct pathways, respond to abiotic stresses in different degree .. and so on. The article must be written in English and must use clear, unambiguous, technically correct text. The article must conform to professional standards of courtesy and expression.
2. Abstract: should have highlights of results and not merely mentioning the methods.
3. Conclusions: there are no results based conclusions. this part needs to be rewritten.

Experimental design

good

Validity of the findings

The expression levels of 9 randomly selected SbKCS genes has not been properly discussed in lieu of other findings. This is an important part of the study. Authors should rewrite with the data in figures/tables.

Additional comments

The article must be written in English and must use clear, unambiguous, technically correct text. The article must conform to professional standards of courtesy and expression.

---

## Round 0.3 · accepted · Accept

The article now seems ok for publication.

·

Basic reporting

Authors have revised their manuscript well as per the comments.

Experimental design

Good

Validity of the findings

Good and appropriate for the study

Additional comments

None